# Hybrid Seq2Seq Architecture for 3D Co-Speech Gesture Generation

KHALED SALEH, University of Technology Sydney (UTS), Australia

This paper describes the co-speech gesture generation system developed by DSI team for the GENEA challenge 2022. The proposed framework features a unique hybrid encoder-decoder architecture based on transformer networks and recurrent neural networks. The proposed framework has been trained using only the official training data split of the challenge and its performance has been evaluated on the testing split. The framework has achieved promising results on both the subjective (specially the human-likeness) and objective evaluation metrics.

ACM Reference Format:

Khaled Saleh. 2022. Hybrid Seq2Seq Architecture for 3D Co-Speech Gesture Generation. In *INTERNATIONAL CONFERENCE ON MULTIMODAL INTERACTION (ICMI '22), November 7–11, 2022, Bengaluru, India.* ACM, New York, NY, USA, 9 pages. https://doi.org/10.1145/3536221.3558064

## 1 INTRODUCTION

The speech-driven gesture generation problem has got some momentum over the past few years. The reason for that is its prevalence in a number of intersecting domains such as human-robot/computer interactions, where body language gestures (also known as co-speech gestures) plays a great role in enhancing the communication skills [6, 18] of social humanoids in physical spaces [19] and virtual avatars in a metaverse [13]. In the literature, the problem of speech-driven gesture generation is often tackled using three main categories of approaches, namely audio-based approaches, transcripts-based approaches and hybrid audio/transcripts-based approaches. As the name implies, the audio-based approaches, are only relying on the audio signals (raw/pre-processed) of the speech to synthesis gestures or body motion, while the transcripts-based approaches utilise only the corresponding transcripts of the speech for the gestures generation. The hybrid audio/transcripts-based approaches, on the other hand, rely on the fusion of both the audio signals and the transcripts in order to generate body gestures.

In this work, we are proposing an audio-based approach, and we formulate the co-speech gestures problem as a sequence-to-sequence (seq2seq) task, where given a long-term speech sequence we auto-regressively predict a long-term sequence of gesticulated motion of full-human body in 3D. Unlike, other audio-based approaches [3, 5, 10], our proposed framework has a hybrid encoder-decoder architecture inspired by the recent work in [7], where the encoder part is based on transformer networks architecture [20], which we rely on its self-attention mechanism to better capture the acoustics of speech such as intonation, prosody, and loudness which are closely correlated to affective gesticulation [17]. The decoder part, on the other hand is based on recurrent neural networks (more specifically LSTM architecture) which we utilise its powerful temporal modelling property to auto-regressively generate a consistent gesture motion.

The rest of the paper will be organised as follows. In Section 2, we will give a brief review of the related work from the literature. Then, in Section 3, we will describe the details of our proposed speech-driven gesture generation system and the data preparation and pre-processing steps we have performed. Later, in Section 4, we will provide

detailed description of the evaluation metrics utilised to assess the performance of our approach. Finally, in Section 5, we conclude our paper and provide directions for potential future works.

## 2 RELATED WORK

The problem of co-speech gestures generation has been commonly tackled using three broad classes of approaches: audio-based approaches [3, 5, 10], transcripts-based approaches [8, 17, 22] and hybrid audio/transcripts-based approaches [1, 11]. For audio-based approaches, Hasegawa et al. [5] introduced one of the early works that relied on recurrent neural networks model (specifically the Bi-Directional Long Short-Term Memory architecture (Bi-Directional LSTM) [4]) for continuous co-speech gestures generation as a sequence of full-body joint positions in 3D. The input to their model was a sequence of extracted audio features via Mel-Frequency Cepstral Coefficients (MFCC) [2]. Additionally, they have utilised a post-processing temporal filtering on the output from the Bi-Directional LSTM model in order to get a smooth sequence of 3D joint positions of the full-body. Similarly, in [10], another recurrent neural networks based model was proposed, however they utilised more higher representations of the motion/speech via the denoising autoencoders [21], and also they get rid of the post-processing filtering step utilised in [5].

For the transcripts-based approaches, Ishi et al. [8], proposed a hand gesture generation model based on the transcripts of speech that was mapped on a word-level into concepts via English language lexical database, WordNet [16]. The extracted concepts are further mapped to two discrete gesture functions that are lastly clustered to generate hand gestures in 3D. In [22], another transcripts-based approach was proposed which was an encode-decoder model, where the encoder was based on Bi-Directional GRU architecture that takes an input transcripts word by word. The encoded features are passed to the decoder which was another recurrent neural network, specifically GRU that generates a sequence of upper-body joint positions in 2D. Finally, for hybrid audio/transcripts-based approaches, one of the early works was introduced in [1] where a deep conditional neural field model that takes as an input a combination of utterance transcription and audio features of the speech to predict a sequence of gestural signs from a pre-defined gestural sign dictionary. More recently, Kucherenko et al. [11] proposed a fully-connected neural network that takes as an input extracted features of speech (log-power mel-spectrogram) and semantic text features extracted via BERT model [9], which in returns predicts the gesture motion as an exponential map representation of upper-body's joint angle rotations.

## 3 DSI SPEECH-DRIVEN GESTURE GENERATION SYSTEM

In our formulation for the co-speech gesture generation problem, our assumption is the availability of a dataset $D = \{(A_i, M_i)\}_{i=1}^N$, where $A = \{a_t\}_{t=1}^n$ is a speech audio sequence with a corresponding audio/acoustic features vector $a_t$, and $M = \{m_t\}_{t=1}^n$ is a sequence of a full-body's gesture motion with $m_t$ aligned to $a_t$. The objective is to reach to a generation model $g(\cdot)$ from $D$, where given an unseen speech audio sequence $A$, a full-body's gesture motion $M$ can be generated based on $g(A)$.

The operation of our seq2seq generation framework (shown in Fig. 1) $g(\cdot)$, will be governed by the joint of and end-to-end two main modules. The first module is the multi-head attention Transformer encoder which will be responsible for transforming the input $A$ into a latent sequence $Z = (z_1, \ldots, z_n)$ $\left(z_i \in \mathbb{R}^{d_z}\right)$, which will be fed to our second module and the LSTM autoregressive decoder, which as the name implies will autoregressively predict a sequence of full-body's gesture motion $M = (m_1, \ldots, m_n)$ in 3D conditioned on $Z$. During the training of our proposed framework, we adopted a curriculum learning strategy similar to the one presented in [7] to overcome the problem of error accumulation which

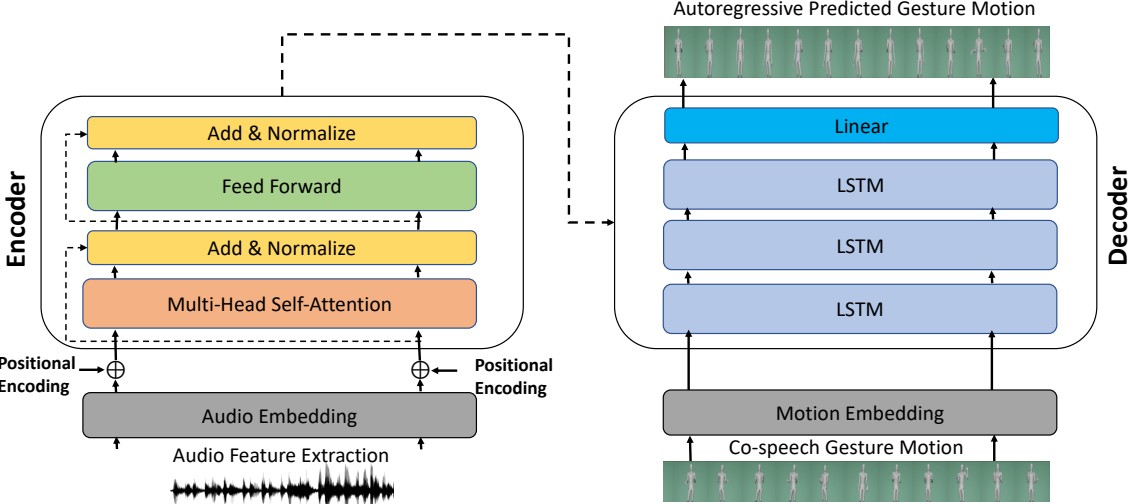

Fig. 1. Our proposed seq2seq framework with hybrid encoder-decoder architecture for co-speech gesture generation.

is commonly associated with autoregressive models. This problem arises due to the fact that autoregressive models during the training phase are not exposed to the prediction errors produced by itself as they are optimised using the ground-truth data (i.e. using teacher-forcing training paradigm).

Thus, in our curriculum learning strategy we address this problem by dynamically during the training process alternate between the fully guided teacher-forcing paradigm and another less guided autoregressive paradigm by utilising the generated motion gestures instead. In the following, we will describe the details of the building blocks of our proposed seq2seq framework. Then we will explain the steps we have followed for the data preparation and processing before feeding them to our framework.

### 3.1 Multi-Head Attention Transformer Encoder

The first module of our proposed framework is the encoder, which takes as an input a sequence of audio features (will be discussed in Section 3.3.1) extracted from the raw audio of the speech. The core of our encoder's architecture is based on the encoder architecture of the original transformer networks first introduced in [20]. One of the main advantages of such architecture is the utilisation of the multi-head self-attention mechanism which can effectively capture the hierarchical representations of the input speech acoustic features. Similar to [20], the input to our encoder is first passed through an embedding layer which linearly transforms the input audio features via learnable weight matrices. As it can be shown at the right hand side of Fig. 1, the output from the embedding layer is added to the positional encoding layer which can be viewed as a way for time-stamping input sequence as transformer networks doe not have the implicit notion of order like LSTM. The positional encoding is done via a wide spectrum of frequencies of sine/cosine as it was formulated in [20]. The encoder itself internally is consisted of (feed-forward neural networks and multi-head self attention) blocks. Additionally, each block is interleaved with a residual connections and a normalisation operation. The multi-head self-attention, or the multi-scaled dot-product attention, works based on the mapping between the so-called 'query' vectors and the pair (key, value) vectors. The dimension of query and key vectors is $d_k$ , where the values vector dimension is $d_v$. The attention operation itself is computed by taking the dot-product between the query

and key vectors divided by the square root of $d_k$ before finally passing them to the softmax function to get their weights by their values. Since scaled dot-product attention operation is done multiple times, the queries, keys and values vectors are extended into matrices $Q, K, V$ respectively. The following formula is the description of how the scaled dot product attention operation is calculated:

$$\text{Attention}(Q, K, V) = \text{softmax}\left(\frac{QK^T}{\sqrt{d_k}}\right)V \tag{1}$$

## 3.2 LSTM Autoregressive Decoder

The decoder module of our proposed framework takes as an input both the latent sequence features $Z$ from the encoder module and during the training phase, the ground-truth co-speech gesture motion of the full-body in 3D. For our decoder architecture, we chose to utilise the LSTM architecture as it was shown recently to be quite effective when it comes to capturing the spatial-temporal dependency between consecutive gesture motion [14]. In total, we have three LSTM layers in addition to one linear layer as part of our decoder. Similar to the encoder, the ground-truth gesture motion (during training phase only), is passed though an embedding layer before they are passed to the first LSTM layer of our decoder.

## 3.3 Dataset

In order to train, validate and test the performance of our proposed framework, we utilised the dataset from the 2022 version of the GENEA challenge [23]. The challenge's dataset is a subset version adapted from [12]. The original dataset are recordings of dyadic interactions between different speakers. In the challenge's dataset, each dyad has been separated into two independent sides with one speaker each. Besides the raw audio recordings of each speaker's speech, the dataset has both word-level time-aligned text transcriptions and time-aligned 3D full-body motion-capture data in BVH format that was captured at 30 Hz. The dataset has been split into three different splits, namely training, validation and testing splits. For the training split, there is a total of 292 speech recordings with a duration that ranges from one to nine minutes. For the validation split, there is a total of 39 speech recordings with average duration around one minute. The testing split on the other hand has total of 40 speech recordings with a duration of one minute each.

In the following, we will describe the preparation and pre-processing steps we have performed on the dataset during the training of our proposed framework.

### 3.3.1 Data Preparation and Pre-processing.
Since the duration for each speech recording is variable, the first preparation step we started with is segmenting each recording into smaller chunks to facilitate that the training process. Instead of doing this segmentation using a sliding window style (which might cut in the middle of a speaking word), we utilised the transcripts that were provided as part of the challenge to roughly segment the starting and ending times of sentences. Since the transcripts provided were on word-level rather than sentence-level, so we heuristically have constructed a sentence based on joining consecutive words until the difference between the starting time of current word and the ending time of the preceding word is more than a threshold value (which in our case was empirically chosen to be 0.5 second). We further, filtered out long sentences to have all sentences under 30 seconds each for an efficient training.

*3.3.2* **Audio Feature Extraction**. For the features extraction from the raw audio recordings, we utilised Librosa library [15] which is commonly used for audio and signal analysis. The set of features we extracted are as follow: mel frequency cepstral coefficients (MFCC) (20-dim) and MFCC delta (20-dim).

## 4 EXPERIMENTAL SETUP AND EVALUATION

### 4.1 Implementation Details

The embedding layer at the start of the encoder of our hybrid encoder-decoder framework has a size of 128 and the input audio features has size $d_a$ of 40. Internally, our encoder contains 2 blocks of (fully-connected feed forward layer and multi-head self-attention layer). The number of heads within the self-attention layer is 8 and the fully-connected feed forward layer has a hidden size of 1024. For each head of the multi-head self-attention layer, it has a scaled dot-product layer with $d_k$ and $d_v$ have size of 64. The length of the gesture motion is under 30 seconds which corresponds to a size of 900 (at sampling rate of 30 Hz). The decoder's input embedding layer has a size of 200, while the dimension of the gesture motion data for 56 joints of the full-body including fingers is 672 (since each joint is represented in 3D by 12 elements 'elements of translation and rotation matrices'). The size of hidden units for each LSTM layer within the encoder is 1024 and the size of hidden units for the output linear layer of the decoder is also 672 (i.e. 56×12).

The objective loss function that was used for training our proposed framework is the $L_1$ and we utilised the Adam optimiser to minimise it with a learning rate of 1e-3 for 500 epochs on NVIDIA GeForce GTX 1080 GPU.

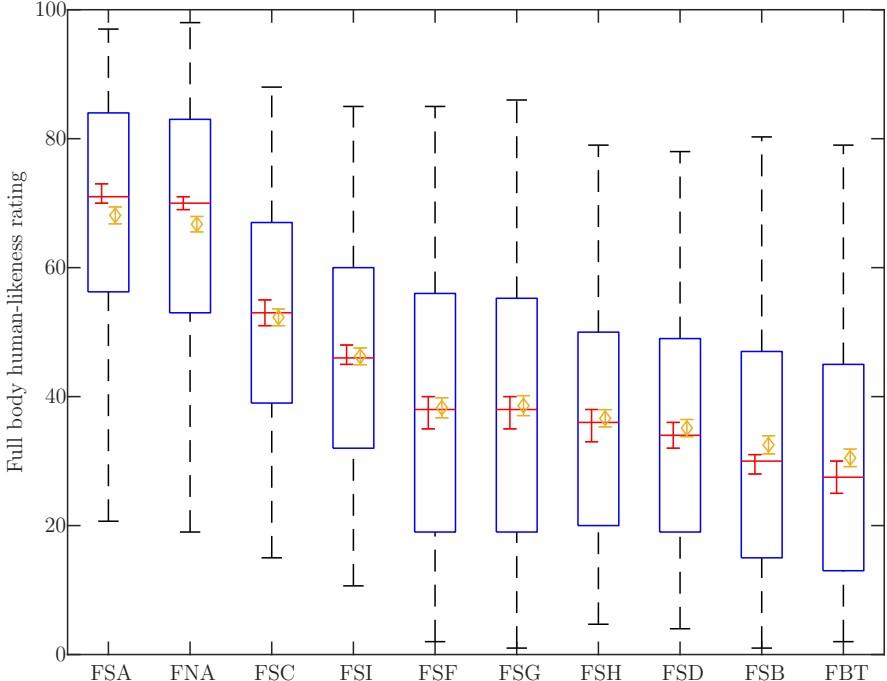

Fig. 2. Box plots visualising the ratings distribution of the human-likeness metric. Red bars are the median ratings (each with a 0.05 confidence interval); yellow diamonds are mean ratings (also with a 0.05 confidence interval). Box edges are at 25 and 75 percentiles, while whiskers cover 95% of all ratings for each condition. Conditions are ordered descending by sample median.

## 4.2 Subjective Evaluation Metrics

The main evaluation metrics for the 2022 GENEA challenge is subjective and is done via the crowd-sourcing platform, Prolific. In specific two main subjective evaluation metrics were studied, namely Human-likeness and Appropriateness.

In the following, the reported results of our system (**FSF**) in comparison to other 7 participants (in addition to the ground-truth natural system, FNA and the baseline text-based approach, FBT [22]) in the full-body tier of the 2022 GENEA challenge according to the aforementioned two evaluation metrics will be presented.

- **Human-likeness:** The Human-likeness metric measures whether the motion of the virtual character looks like the motion of a real human without hearing its corresponding speech.
- **Appropriateness:** On the other hand, the appropriateness metric measures whether the motion of the virtual character is appropriate for the given speech, controlling for the humanlikeness of the motion. This metric can be also referred to as "specificity".

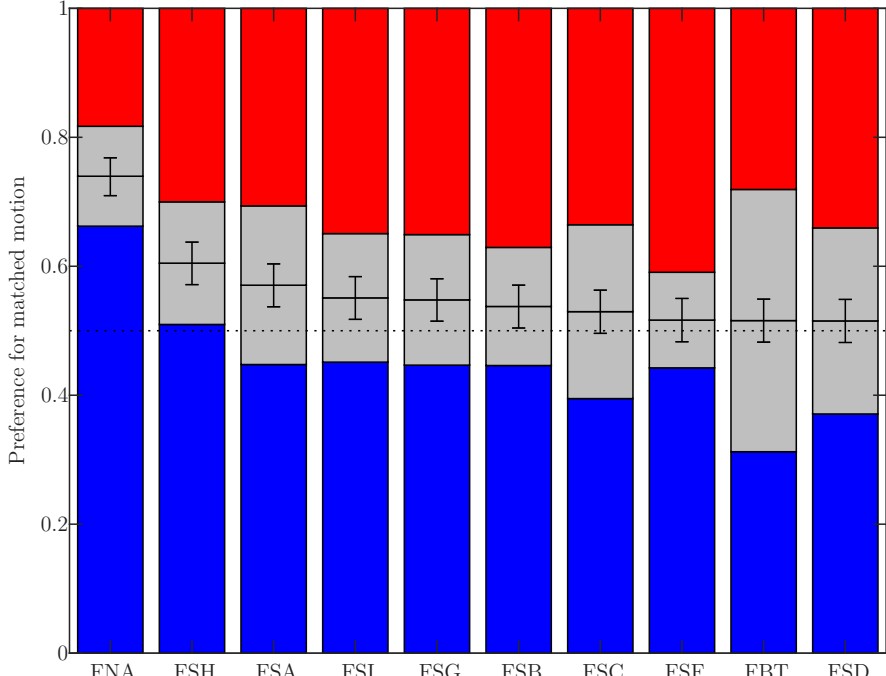

Fig. 3. Bar plots visualising the response distribution of the appropriateness metric. The blue bar (bottom) represents responses where subjects preferred the matched motion, the light grey bar (middle) represents tied ("They are equal") responses, and the red bar (top) represents responses preferring mismatched motion, with the height of each bar being proportional to the fraction of responses in each category. The black horizontal line bisecting the light grey bar shows the proportion of matched responses after splitting ties, each with a 0.05 confidence interval. The dashed black line indicates chance-level performance. Conditions are ordered by descending preference for matched after splitting ties.

## 4.3 Objective Evaluation Metrics

In order to evaluate the performance of our proposed framework ((FSF)) quantitatively in comparison to the other participant systems of the challenge, we utilised the following three evaluation metrics:

*4.3.1* ***Average jerk****.* This metric is commonly used to measure motion smoothness. A perfectly natural system should have average jerk very similar to natural motion

*4.3.2* ***Comparing speed histograms****.* This metric is used evaluate gesture quality. Since well-trained models should produce motion with similar properties to that of the actor it was trained on. In particular, it should have a similar motion-speed profile for any given joint. This metric is calculated using the Hellinger distance.

*4.3.3* ***Canonical correlation analysis (CCA)****.* This metric is used to find a sequence of linear transformations of each variable set, such that the correlations between the transformed variables are maximised. Based on this correlation, it can be utilised as a similarity measure.

### 4.4 Results and Discussions

In Fig. 2 and 3, the reported results for the human-likeness and appropriateness are shown respectively. For the human-likeness, it can be notice that our system ((FSF)) was one of the top 4 systems that have achieved the highest levels of human-likeness. It can be noticed also that the crowd-sourced subjects were able to distinguish between the real human motion (FNA) and the rest of the systems generated motion (with the only one exception of FSA). On the

| Condition | Average jerk | Average acceleration | Global CCA | Hellinger distance average |
|---|---|---|---|---|
| FNA | ***31324.43 +- 6588.19*** | ***797.53 +- 207.71*** | *1* | *0* |
| **FBT** | 3504.05 +- 1089.93 | 177.31 +- 56.01 | 0.73848 | 0.2670497593 |
| **FSA** | 14598.06 +- 2970.64 | **668.44 +- 160.97** | 0.84948 | **0.04096112013** |
| **FSB** | **27160.94 +- 4679.38** | **628.07 +- 115.74** | 0.78182 | **0.04952276147** |
| **FSC** | 5129.45 +- 2116.81 | 332.25 +- 129.43 | 0.81826 | 0.1252592381 |
| FSD | 8691.69 +- 8317.16 | 405.42 +- 256.97 | 0.88646 | 0.1323802021 |
| FSF | **22628.91 +- 6241.05** | **666.02 +- 223.34** | **0.91574** | 0.1945411681 |
| FSG | 5564.40 +- 2383.01 | 282.23 +- 127.24 | **0.99154** | **0.05967038642** |
| FSH | 8632.45 +- 2436.03 | 312.80 +- 92.41 | **0.968** | 0.1036145704 |
| **FSI** | 7373.95 +- 1711.13 | 345.36 +- 97.74 | 0.78944 | 0.1106242683 |

Fig. 4. Objective evaluation results.

other hand, for the appropriateness metric, our proposed framework seems to be lagging in this metric despite the fact that it's on par with the other participant systems when it comes to the preference of the subjects to the matched motion.

Regarding the objective measures, in Fig. 4, we can notice that our system (**FSF**) is one of top 2 systems that are closer to the natural motion system (FNA) in terms of average jerk/accelerations and one of the top 3 systems when it comes to Global CCA score. Based on those scores, we can notice that our system has done a good job in capturing the different styles of motion of the actors of in the dataset of the 2022 GENEA challenge (as it can be shown from the qualitative results here ).

## 5 CONCLUSION

In this work we has proposed an innovative hybrid encoder-decoder framework that can effectively generate co-speech gestures that can better capture the gesticulation style of different speaks. The performance of framework has been evaluated subjectively and objectively. On the subjective front, it was one of the top 4 participant systems in the 2022 GENEA challenge when it comes to the human-likeness. On the objective front, it was one of the top 2 performing systems that have similar jerk and acceleration profiles to the natural motion system.

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
