# OpenReview forum: "Hybrid Seq2Seq Architecture for 3D Co-Speech Gesture Generation"
_ACM.org/ICMI/2022/Workshop/GENEA — GENEA Challenge & Workshop 2022 Mainproceeding_

### Official Review · Reviewer_tdQF · 2022-07-30
**Some clarification required.**

**Rating:** 7
**Confidence:** 3

**Review:**

Overall, the presented approach is quite interesting. However, some clarification will strengthen the paper. First of all, it is not entirely clear how the decorer works. Is it replicate the pipeline of Huang et al. 2020 - does it generate motion frame by frame feeding previously generated poses (in a non teacher-forcing case)? By Figure 1 it  looks like it generates the whole sequence from given input. Also, it is not clear how output features of decoder (Z) are incorporated with input poses. Are they concatenated before the first LSTM layer or there is another strategy?

Furthermore,  there are several questions that are interesting to discuss. As mentioned in the data preparation section, only gestures relevant to actual speech were used. It is interesting how the proposed model works in cases when the target person is silent and listening to the conversation partner. Second, it is interesting how much GPU memory is utilized by self-attention for such long sequences? For example, in Huang et al. 2020 the local self-attention was used to optimize memory consumption. Finally, some observations on generated motion would be also useful. What could be improved?

I hope the comments above would help you to strengthen the paper and clarify some aspects. Also, in my opinion, the word "innovative" is not very suitable in this case, since it is very similar to  Huang et al. 2020.

---

### Official Review · Reviewer_hoQP · 2022-08-07
**Combining a transformer-based encoder and a RNN-based decoder paid off with nice performance.**

**Rating:** 6
**Confidence:** 3

**Review:**

The paper describes a novel attempt to combine a transformer-based encoder and a RNN-based decoder to build a hybrid model for audio-to-gesture generation, which showed promising performance with rankings in subjective evaluations higher than average.

The proposed model is clearly presented in the paper with details. Presenting a novel way of fusing different models with experimental results is a nice contribution to the challenge.

Some comments to improve the paper:
1) According to the authors, one of the rationale behind using a transformer-based encoder is to better capture the para-verbal features like intonation, prosody and loudness. But it is not explained in the paper how such para-verbal features contribute to the performance of gesture generation. Some explanations would help better understand the benefits of the proposed model.
2) In section 3, the authors mention that curriculum learning is employed to overcome the problem of error accumulation. More details would be helpful for the readers.
3) In conclusion, it is mentioned that the proposed hybrid encoder-decoder model could generate gestures that better capture the individual styles of different speakers. Any quantitative or qualitative results that support the comments would be interesting because modeling personalized gesticulation style could be one of the ultimate goals of gesture generation technology.

---

### Official Review · Reviewer_suXb · 2022-08-09
**A combined transformer and RNN for gesture generation**

**Rating:** 6
**Confidence:** 3

**Review:**

Paper strengths:
- Combining a transformer encoder and an LSTM decoder is interesting.
- In objective evaluation, the proposed system produces gesture motions close to the natural motion in terms of the average jerk, acceleration, and global CCA.

Comments and questions:
- While the authors do not clearly state, not only the curriculum learning strategy but also the model architecture seems to be heavily inspired by [17], i.e., transformer encoder + LSTM decoder. Therefore, an adequate reference and description should be added in Sec. 3.
- Lines 325-326: “our system was one of the top 4 systems that have achieved the highest levels of human-likeness”: For me, this sentence sounds a bit overrated. In Fig. 2, the difference between FSF (“our system”) and FSG is marginal.
- Lines 352-353: “our system is … one of the top 3 systems when it comes to Global CCA and Hellinger distance scores”: However, in terms of Hellinger distance, the lower, the closer to the natural motion. Thus, FSF is the 8th in the Hellinger distance evaluation.

Minor points:
- In Fig. 1, the images of the virtual agent are unnaturally stretched. The aspect ratio should be fixed.
- Line 223: l_1 -> L_1
- Sec. 4.2.1 and 4.2.2 are too short to be separate sub-subsection. It would be better to use \noindent{\textbf{Human-likeness.}} instead.
- Line 311: motion -> motion. (add period)

Overall:

Adapting a dance generation model to gesture generation is promising, and the proposed system produces gesture motions having close properties to the natural motion in some objective evaluation metrics. On the other hand, the paper seems to be a bit exaggerated in terms of the novelty of the model architecture and evaluation results.

---

### Decision · Program_Chairs · 2022-08-11

**Decision:**

Accept (Main proceeding)

**Comment:**

Congratulations! All the reviewers recommended accepting this paper for the <>. They appreciate the fusion of a transformer encoder with an LSTM decoder, and the presentation of the results.

We suggest that the authors carefully consider the feedback received from the reviewers and use it to improve their manuscript for the challenge camera-ready submission deadline. Below follows some input from the chairs, based on the paper and the reviews:

Per reviewer comments, please provide some more detail on why this approach is better in capturing para-verbal features.
The manuscript makes a strong claim in saying that the model is good at capturing individual styles. If this claim is retained, it would be preferred to support the claim with additional data.